# Infection-related severe maternal outcomes and case fatality rates in 43 low and middle-income countries across the WHO regions: Results from the Global Maternal Sepsis Study (GLOSS)

**Adama Baguiya**[1,2]*, **Mercedes Bonet**[3], **Vanessa Brizuela**[3], **Cristina Cuesta**[4], **Marian Knight**[5], **Pisake Lumbiganon**[6], **Edgardo Abalos**[7], **Séni Kouanda**[1,2], **WHO Global Maternal Sepsis Study Research Group**¶

1 Biomedical and Public Health Department, Institut de Recherche en Sciences de la Santé (IRSS), Ouagadougou, Burkina Faso, 2 Doctoral School, Saint Thomas d'Aquin University, Ouagadougou, Burkina Faso, 3 Department of Reproductive Health and Research, UNDP/UNFPA/UNICEF/WHO/World Bank Special Programme of Research, Development and Research Training in Human Reproduction (HRP), World Health Organization, Geneva, Switzerland, 4 Faculty of Economics and Statistics, National University of Rosario, Rosario, Argentina, 5 Nuffield Department of Population Health, National Perinatal Epidemiology Unit, University of Oxford, Oxford, United Kingdom, 6 Faculty of Medicine, Department of Obstetrics and Gynaecology, Khon Kaen University, Khon Kaen, Thailand, 7 Centro Rosarino de Estudios Perinatales (CREP), Rosario, Argentina

¶ Membership of WHO Global Maternal Sepsis Study Research Group is provided in S1 Acknowledgments
* abaguiya@gmail.com

**Data Availability Statement:** The data used for this analysis can be made available upon request.

## Abstract

The highest toll of maternal mortality due to infections is reported in low and middle-income countries (LMICs). However, more evidence is needed to understand the differences in infection-related severe maternal outcomes (SMO) and fatality rates across the WHO regions. This study aimed to compare the burden of infection-related SMO and case fatality rates across the WHO regions using the Global Maternal Sepsis Study (GLOSS) data. GLOSS was a hospital-based one-week inception prospective cohort study of pregnant or recently pregnant women admitted with suspected or confirmed infection in 2017. Four hundred and eight (408) hospitals from 43 LMICs in the six WHO regions were considered in this analysis. We used a logistic regression model to compare the odds of infection-related SMOs by region. We then calculated the fatality rate as the proportion of deaths over the total number of SMOs, defined as maternal deaths and near-misses. The proportion of SMO was 19.6% (n = 141) in Africa, compared to 18%(n = 22), 15.9%(n = 50), 14.7%(n = 48), 12.1%(n = 95), and 10.8%(n = 21) in the Western Pacific, European, Eastern Mediterranean, Americas, and South-Eastern Asian regions, respectively. Women in Africa were more likely to experience SMO than those in the Americas (aOR = 2.41, 95%CI: [1.78 to 2.83]), in South-East Asia (aOR = 2.60, 95%CI: [1.57 to 4.32]), and the Eastern Mediterranean region (aOR = 1.58, 95%CI: [1.08 to 2.32]). The case fatality rate was 14.3%[3.05% to 36.34%] (n/N = 3/21) and 11.4%[6.63% to 17.77%] (n/N = 16/141) in the South-East Asia and Africa, respectively. Infection-related SMOs and case fatality rates were highest in

Contact of the GLOSS coordinator bonetm@who.int.

**Funding:** AB received funding from the HRP Alliance, part of the UNDP-UNFPA-UNICEF-WHO-World Bank Special Programme of Research, Development, and Training in Human Reproduction (HRP), a cosponsored program executed by the World Health Organization (WHO), to complete his studies. The main study was financially supported by the UNDP–UNFPA–UNICEF–WHO–World Bank Special Programme of Research, Development and Research Training in Human Reproduction, Department of Sexual and Reproductive Health and Research, WHO, Geneva, Switzerland (project A65787), Merck Sharp & Dohme, a wholly owned subsidiary of Merck (Kenilworth, NJ, USA), through its Merck for Mothers programme, and the United States Agency for International Development (grant GHA-G-00-09-00003). The funders had no role in study design, data collection and analysis, decision to publish, or manuscript preparation. The named authors alone are responsible for the views expressed in this publication and do not necessarily represent the decisions or the policies of the UNDP-UNFPA-UNICEF-WHO-World Bank Special Programme of Research, Development and Research Training in Human Reproduction (HRP) or the World Health Organization (WHO) or the other affiliated institutions.

**Competing interests:** The authors have declared that no competing interests exist.

Africa and Southeast Asia. Specific attention and actions are needed to prevent infection-related maternal deaths and severe morbidity in these two regions.

## Introduction

Maternal sepsis is defined as "*a life-threatening condition defined as organ dysfunction resulting from infection during pregnancy, childbirth, post-abortion, or post-partum period*" [1]. It can complicate all types of infection, including pregnancy-related ones, and lead to severe maternal outcomes. The incidence of direct obstetric infections in late pregnancy and the postpartum period in sub-Saharan Africa is 3.5% and 2.4%, respectively. In South Asia and sub-Saharan Africa, women with infection in the postpartum period are 1.8 times more likely to die compared to those without infection [2]. Globally, direct obstetric infections represent the third cause of maternal mortality. They are responsible for almost 11% of all maternal deaths worldwide; specific to sub-Saharan Africa and South Asia, these numbers rise to 12% and 14%, respectively [3,4].

Previous publications from the Global Maternal Sepsis Study (GLOSS) indicated that for every 1000 live births to women with suspected or confirmed infection, 11 women experience infection-related severe maternal outcomes (SMO), which include maternal near-misses and deaths. Out of those SMOs, the global case fatality rate is 6.8%. The highest rates and proportions are observed in low-income countries, with SMO in 15 women per 1000 live births and a 15% fatality rate among infection-related SMOs [5].

There is a growing literature on maternal infection and sepsis, particularly with country-specific data in high- [6–8], low, and middle-income countries [9–12]. Previous comparisons across regions or income levels used mathematical models such as the Global Burden of Diseases [13–15] or systematic reviews [16]. However, they were often limited by insufficient data in resource-constrained settings. Nevertheless, these studies highlighted inequalities related to the burden of maternal infections and hospital readiness for their identification and treatment between low- and high-income countries and across levels of sociodemographic indexes [5,14,16,17]. Based on the high burden of all-cause maternal mortality in sub-Saharan Africa (70% of the global burden) [18], we hypothesized that there could be disparities across the WHO regions, and women could be more likely to experience severe morbidity or die from maternal infections in Africa than in the other regions.

The maternal mortality ratio is an indicator of poor quality of care provided to women during and after pregnancy [19]; hence, the provision of optimal care for maternal and neonatal infection is part of the priorities in maternal and child health service delivery [20]. When systems fail to prevent maternal infections, the capacity of facilities to respond becomes crucial for women's survival. Country and region-level data on SMO could be helpful for policymakers and health systems managers to understand and assess hospitals' performance in caring for women with infection during pregnancy or in the postpartum period [21]. Therefore, GLOSS data give a unique opportunity to explore the regional disparities in hospitals' capacity to identify and manage maternal infections and assess the burden of infection-related SMOs among LMICs in each region. In this study, we aimed to describe the hospitals' capacity for identifying and managing maternal infections and to compare the proportion of infection-related SMOs and their associated case fatality rate in the African region versus the other regions based on the WHO classification of the countries, using GLOSS data.

## Materials and methods

### Study design and setting

The GLOSS was a hospital-based one-week inception prospective cohort study. Forty-three (43) LMICs (out of 52 countries involved in the entire GLOSS) were included in this analysis.

There were 13 countries in the African Region (AFR) and five in Europe. The other four regions were the Americas (AMR: 10 countries), South-East Asia (SEAR: 5 countries), Eastern Mediterranean (EMR: 6 countries), and the Western Pacific (WPR: 4 countries) [12]. Nine (09) high-income countries of the GLOSS, including eight (08) in Europe [22] and one (01) in the Americas, were excluded from this analysis.

The GLOSS identified and included all women with confirmed or suspected infection during any stage of pregnancy and up to the 42nd day after the end of the pregnancy who were admitted or already hospitalized for at least 12 hours in participating health facilities.

Women were enrolled between November 28 and December 4, 2017, in geographical areas within 52 purposively selected countries in all six WHO regions. The study procedures are described elsewhere [12].

### Data sources

Briefly, data were collected on the characteristics of the study's geographical areas, hospitals, and individual participants. Information on the geographical area's characteristics, including its population, was captured at the area level. At the hospital level, data were collected on the hospital size, level of care, type (public or private), the number of deliveries in the year before the study, and staff availability by category. It also included a set of equipment and services, among others. Individual-level data were extracted from patients' records by health workers in the participating hospitals. Individual information included women's sociodemographics, obstetric background, clinical profile, pregnancy outcomes, and clinical management data. All procedures and diagnoses were based on the staff's routine clinical activities. Data collectors did not have to interact with the patients except when seeking consent. No additional resources, guidelines, or interventions were provided for the study, except that GLOSS was accompanied by an awareness campaign fully described elsewhere [23].

### Variables and measurement

The primary outcome variable in this analysis is an infection-related SMO, including infection-related maternal death and near-miss. Maternal deaths were either directly linked to the infection or had an infection as a contributing cause. For maternal near-miss, we applied the WHO definition: a woman who nearly died but survived a life-threatening condition during pregnancy, childbirth, or postpartum or post-abortion period [24]. Participants were classified as near-miss cases if they had at least one of the WHO near-miss criteria [25]. The independent variable was the region. Participants were classified according to the region to which their country belonged. Six categories represented the WHO regions: AFR, AMR, EMR, EUR, SEAR, and WPR [26].

Data on the hospital capacity were collected using a dedicated form, which gathered information on the availability of medical and laboratory equipment, drugs, staff, and infection prevention and control procedures and protocols in the facility. We used the variables related to the country's level of income (upper middle, lower middle, and low income), hospitals' location (urban or rural/peri-urban), level of care (primary, secondary, or tertiary), type of facility (public or private/non-governmental organization/faith-based), capacity to provide Emergency Obstetric and Newborn Care (basic -BEmONC-, comprehensive -CEmONC-, or

neither), affiliation to a university or not, facility size (number of births the previous, year with, 2016, with four categories: <1000 livebirths, 1000–2499 livebirths, 2500–4499 livebirths, and ≥4500 livebirths), staff availability 24/7 (either physically or on call), availability of infection prevention and control (IPC) committee (yes or no), of a surveillance system for antibiotics and antimicrobials (yes or no), and a training or continuing education system on IPC and hospital hygiene (yes or no). In addition, we computed three composite variables related to i) availability of services and equipment, ii) identification capacity, and iii) management capacity. For each of these variables, we used a set of dummy variables presented in **S1 Table** to create an index (score) for each hospital using a Principal Component Analysis (PCA). Hospitals were then ranged and divided into three groups of equal size (each group contains one-third of the hospitals), namely the lowest (coded 1), medium (coded 2), and highest (coded 3) level of hospital service availability, capacity, or management index [27].

### Data analysis

We used descriptive statistics to present data on the facility's general characteristics, including the availability of services and equipment, identification, and management capacity.

We then calculated the proportion of women who had SMO among all women with infection and the rate of infection-related SMO per 1000 live births with their 95% confidence intervals (95%CI), stratified by region. Cases Fatality Rates by region were computed as the proportion of deaths due to infections over the total number of women with SMO.

Finally, we fitted a logistic regression model to assess the association between the WHO region and the occurrence of SMO. We computed crude and adjusted odds ratios (aOR) with their 95%CI using a backward stepwise approach with facility-level characteristics. The final model was adjusted based on facility size, location, capacity to manage maternal infections, and the availability of a routine training program on infection management. We used the Akaike Information Criteria (AIC) and the "*linktest*" command to compare the models and assess the goodness of fit. All analyses were performed using Stata 18.

### Ethical considerations

During the GLOSS, written informed consent or waiver of consent was obtained to extract data from the patient's records (based on the requirements of each local institutional or national ethics committee). We also explained to participants that they could opt out of the study at any time and request the withdrawal of their data. All study records, forms, logs, and data were kept confidential. Data were entered with hospitals and participants' sequential numbers. No identifiable information was entered into the system. Data were anonymized for all the analysis. The study protocol was submitted to and approved by the WHO Ethics Review Committee (protocol ID A65787) and the ethics committees of the respective countries and facilities according to national regulations.

## Results

This analysis included 2466 women from 408 facilities in 43 LMICs. Most women were from the Americas (31.2%, n = 788) and Africa (30.2%, n = 718). Overall, 577 women had infections with complications, and 377 had severe maternal outcomes (Table 1).

### Services availability and hospital capacity

Table 2 presents the general characteristics of the facilities, staff, services, and equipment availability and their capacity to identify and manage maternal infections. Most of the facilities

**Table 1. Study participants and distribution of infections, severe cases, and maternal deaths by region.**

| | Overall | African region | Region of the Americas | Eastern Mediterranean region | European region | South-Eastern Asian region | Western Pacific Region |
|---|---|---|---|---|---|---|---|
| Countries | 43 | 13 | 10 | 6 | 5 | 5 | 4 |
| Number of hospitals | 408 | 126 | 88 | 46 | 57 | 35 | 56 |
| Women who had a maternal infection | 2466 | 718 | 788 | 327 | 316 | 195 | 122 |
| *Less severe infection* | *1512* | *426* | *511* | *205* | *179* | *129* | *62* |
| *Infection with complication* | *577* | *151* | *192* | *74* | *87* | *45* | *38* |
| *Infection-related severe maternal outcomes** | *377* | *141* | *95* | *48* | *50* | *21* | *22* |
| Infection-related maternal deaths | 26 | 16 | 3 | 1 | 1 | 3 | 2 |

*Severe maternal outcomes include near-miss as per the WHO definition and maternal deaths.

List of countries

Africa: Benin, Burkina Faso, Cameroon, Ethiopia, Ghana, Kenya, Malawi, Mali, Mozambique, Nigeria, Senegal, South Africa, and Zimbabwe.

Americas: Argentina, Bolivia, Brazil, Colombia, Ecuador, Guatemala, Honduras, Mexico, Nicaragua, and Peru.

Eastern Mediterranean: Afghanistan, Egypt, Lebanon, Morocco, Pakistan; and Sudan.

Europe: Kazakhstan, Kyrgyzstan, Moldova, Romania, and Tajikistan.

South-East Asia: India, Myanmar, Nepal, Sri Lanka, and Thailand.

Western Pacific: Philippines, Mongolia, Viet-Nam, and Cambodia.

were located in urban areas (77.9%, n = 317 in total) and were comprehensive EmONC facilities (80.4%, n = 328), except in the Western Pacific (48.2%, n = 27). One out of four facilities (27.4%, n = 111) had more than 4500 births annually. Regarding staff availability, midwives were available in at least 80% of the facilities (85.6%, n = 107, in Africa), except in the Americas (40.9%, n = 36). But, the availability of internal medicine and infectious disease specialists was lower in Africa (26.6%, n = 33) and higher in the Americas (72.4%, n = 63). In Africa, the antibiotic/antimicrobial use surveillance system was available only in 43.6% (n = 55) of the facilities. In all the other regions, more than 60% of the facilities had a surveillance system. The highest percentage was in the Western Pacific, with 80.4% (n = 45). Regarding the availability of an IPC Committee and a training/continuing education system on IPC/hospital hygiene, the highest percentages were reported in Europe (93.0%, n = 53 and 90.9%, n = 50, respectively) and the lowest in Africa (73%, n = 92 and 76%, n = 95, respectively). In addition, Africa (43.7%, n = 55) and the Western Pacific (55.4%, n = 31) were the two regions where most facilities fell in the group of the lowest level of services and equipment availability, compared to the Americas (8%, n = 7) and South-East Asia (17.1%, n = 6).

The distribution of the individual items related to service availability is presented in S2 Table. More than half of the facilities in the Western Pacific (51.8%, n = 29) and over one-third in Africa (36.5%; n = 46) fell in the category of the lowest level of identification capacity, whereas two-thirds in the Americas (65.9%, n = 58) were in the category of the highest level. The lowest management capacity level was represented by 38.1% (n = 48) in Africa and 10.2% (n = 9) in the Americas. The percentages of the individual items of the capacity of identification and management are presented in S3 and S4 Tables.

## Severe maternal outcome and associated fatality rate

In total, infection-related severe maternal outcomes were experienced by 15.3% (n = 377, 95% CI:13.9% to 16.8%) of the women. Fig 1 shows that Africa had the highest percentage of SMO

**Table 2. Characteristics of the facilities, services, and equipment available in the facility (n = 408).**

| Variable | Region | | | | | | | | | | | | ALL | |
|---|---|---|---|---|---|---|---|---|---|---|---|---|---|---|
| | *African* | | *The Americas* | | *Eastern Mediterranean* | | European | | *South-Eastern Asian* | | Western Pacific | | | |
| | n | % | n | % | n | % | n | % | n | % | n | % | N | % |
| **Location** | | | | | | | | | | | | | | |
| Urban | 95 | *75.4* | 88 | *100.0* | 42 | *91.3* | 50 | *87.7* | 20 | *57.1* | 22 | *40.0* | 317 | *77.9* |
| Rural and peri-urban | 31 | *24.6* | 0 | *0.0* | 4 | *8.7* | 7 | *12.3* | 15 | *42.9* | 33 | *60.0* | 90 | *22.1* |
| **Level of care** | | | | | | | | | | | | | | |
| Primary | 25 | *19.8* | 4 | *4.6* | 15 | *32.6* | 17 | *29.8* | 4 | *11.4* | 5 | *8.9* | 70 | *17.2* |
| Secondary | 59 | *46.8* | 27 | *31.0* | 13 | *28.3* | 24 | *42.1* | 20 | *57.1* | 37 | *66.1* | 180 | *44.2* |
| Tertiary | 42 | *33.3* | 56 | *64.4* | 18 | *39.1* | 16 | *28.1* | 11 | *31.4* | 14 | *25.0* | 157 | *38.6* |
| **Type of facility** | | | | | | | | | | | | | | |
| Public | 89 | *70.6* | 51 | *57.9* | 42 | *91.3* | 52 | *92.9* | 33 | *94.3* | 48 | *85.7* | 315 | *77.4* |
| Private/NGO/Faith-based | 37 | *29.4* | 37 | *42.1* | 4 | *8.7* | 4 | *7.1* | 2 | *5.7* | 8 | *14.3* | 92 | *22.6* |
| **EmONC facility** | | | | | | | | | | | | | | |
| CEmONC | 103 | *81.7* | 80 | *90.9* | 44 | *95.6* | 47 | *82.5* | 27 | *77.1* | 27 | *48.2* | 328 | *80.4* |
| BEmONC | 19 | *15.1* | 7 | *7.9* | 2 | *4.4* | 1 | *1.7* | 6 | *17.1* | 5 | *8.9* | 40 | *9.8* |
| None | 4 | *3.2* | 1 | *1.2* | 0 | *0.0* | 9 | *15.8* | 2 | *5.7* | 24 | *42.9* | 40 | *9.8* |
| **University hospital** | | | | | | | | | | | | | | |
| No | 72 | *57.1* | 23 | *26.1* | 27 | *58.7* | 34 | *63.0* | 24 | *68.6* | 39 | *83.0* | 219 | *55.3* |
| Yes | 54 | *42.9* | 65 | *73.9* | 19 | *41.3* | 20 | *37.0* | 11 | *31.4* | 8 | *17.0* | 177 | *44.7* |
| **Maternity only** | | | | | | | | | | | | | | |
| No | 118 | *93.6* | 80 | *90.9* | 36 | *78.3* | 40 | *70.2* | 20 | *57.1* | 34 | *60.7* | 328 | *80.4* |
| Yes | 8 | *6.4* | 8 | *9.1* | 10 | *21.7* | 17 | *29.8* | 15 | *42.9* | 22 | *39.3* | 80 | *19.6* |
| **Facility size** | | | | | | | | | | | | | | |
| <1000 livebirths | 26 | *20.8* | 18 | *20.7* | 1 | *2.2* | 20 | *35.1* | 14 | *40.0* | 20 | *35.7* | 99 | *24.4* |
| 1000–2499 livebirths | 42 | *33.6* | 22 | *25.3* | 12 | *26.7* | 9 | *15.8* | 6 | *17.1* | 21 | *37.5* | 112 | *27.6* |
| 2500–4499 livebirths | 27 | *21.6* | 28 | *32.2* | 8 | *17.8* | 9 | *15.8* | 4 | *11.4* | 7 | *12.5* | 83 | *20.5* |
| ≥4500 livebirths | 30 | *24.0* | 19 | *21.8* | 24 | *53.3* | 19 | *33.3* | 11 | *31.4* | 8 | *14.3* | 111 | *27.4* |
| **Staff availability 24/7*** | | | | | | | | | | | | | | |
| *Midwife* | 107 | *85.6* | 36 | *40.9* | 38 | *82.6* | 56 | *98.3* | 33 | *94.3* | 47 | *83.9* | 317 | *77.9* |
| *Obstetrics specialist* | 55 | *44.0* | 71 | *80.7* | 18 | *39.1* | 54 | *94.7* | 21 | *60.0* | 31 | *55.4* | 250 | *61.4* |
| *Internal medicine/Infectious disease specialist* | 33 | *26.6* | 63 | *72.4* | 10 | *21.7* | 30 | *53.6* | 19 | *54.3* | 34 | *60.7* | 189 | *46.8* |
| *Anesthesiologist* | 33 | *26.8* | 73 | *83.0* | 24 | *52.2* | 51 | *89.5* | 16 | *45.7* | 28 | *50.0* | 225 | *55.6* |
| Availability of infection Prevention and Control Committee | 92 | *73.0* | 78 | *88.6* | 42 | *91.3* | 53 | *93.0* | 30 | *85.7* | 52 | *92.9* | 347 | *85.1* |
| Availability of surveillance system of antibiotics/antimicrobials use | 55 | *43.6* | 68 | *77.3* | 33 | *71.7* | 42 | *73.7* | 22 | *62.9* | 45 | *80.4* | 265 | *64.9* |
| Availability of a training and/or continuing education system on infection prevention and control/ hospital hygiene | 95 | *76.0* | 75 | *85.2* | 41 | *89.1* | 50 | *90.9* | 28 | *80.0* | 45 | *80.4* | 334 | *82.5* |
| **Services and equipment availability** | | | | | | | | | | | | | | |
| *Lowest availability* | 55 | *43.7* | 7 | *8.0* | 15 | *32.6* | 22 | *38.6* | 6 | *17.1* | 31 | *55.4* | 136 | *33.3* |
| *Medium availability* | 39 | *30.9* | 37 | *42.1* | 16 | *34.8* | 12 | *21.1* | 14 | *40.0* | 14 | *25.0* | 132 | *32.4* |
| *Highest availability* | 32 | *25.4* | 44 | *50.0* | 15 | *32.6* | 23 | *40.4* | 15 | *42.9* | 11 | *19.6* | 140 | *34.3* |
| **Identification capacity** | | | | | | | | | | | | | | |

*(Continued)*

**Table 2.** (Continued)

| Variable | Region | | | | | | | | | | | ALL | |
|---|---|---|---|---|---|---|---|---|---|---|---|---|---|
| | *African* | | *The Americas* | | *Eastern Mediterranean* | | European | | *South-Eastern Asian* | | Western Pacific | | | |
| *Lowest capacity* | 46 | *36.5* | 7 | *8.0* | 21 | *45.6* | 27 | *47.4* | 6 | *17.1* | 29 | *51.8* | 136 | *33.3* |
| *Medium capacity* | 31 | *24.6* | 23 | *26.1* | 17 | *37.0* | 24 | *42.1* | 11 | *31.4* | 16 | *28.6* | 122 | *29.9* |
| *Highest capacity* | 49 | *38.9* | 58 | *65.9* | 8 | *17.4* | 6 | *10.5* | 18 | *51.4* | 11 | *19.6* | 150 | *36.8* |
| **Management capacity** | | | | | | | | | | | | | |
| *Lowest capacity* | 48 | *38.1* | 9 | *10.2* | 20 | *43.5* | 23 | *40.3* | 8 | *22.9* | 27 | *48.2* | 135 | *33.1* |
| *Medium capacity* | 49 | *38.9* | 20 | *22.7* | 17 | *37.0* | 25 | *43.9* | 8 | *22.9* | 18 | *32.1* | 137 | *33.6* |
| *Highest capacity* | 29 | *23.0* | 59 | *67.1* | 9 | *19.6* | 9 | *15.8* | 19 | *54.3* | 11 | *19.6* | 136 | *33.3* |

*Available 24/7 in the hospital or on a call.

(19.6%, 95%CI:16.8% to 22.7%), followed by the Western Pacific (18%, 95%CI:11.7% to 26%). The confidence intervals presented in S1 Fig indicate that the percentage in Africa was statistically higher than the ones in the Americas (12.1%, 95%CI: 9.9 to 14.5) and South-East Asia (10.7%, 95%CI: 6.8 to 16.0), where the lowest percentages were reported.

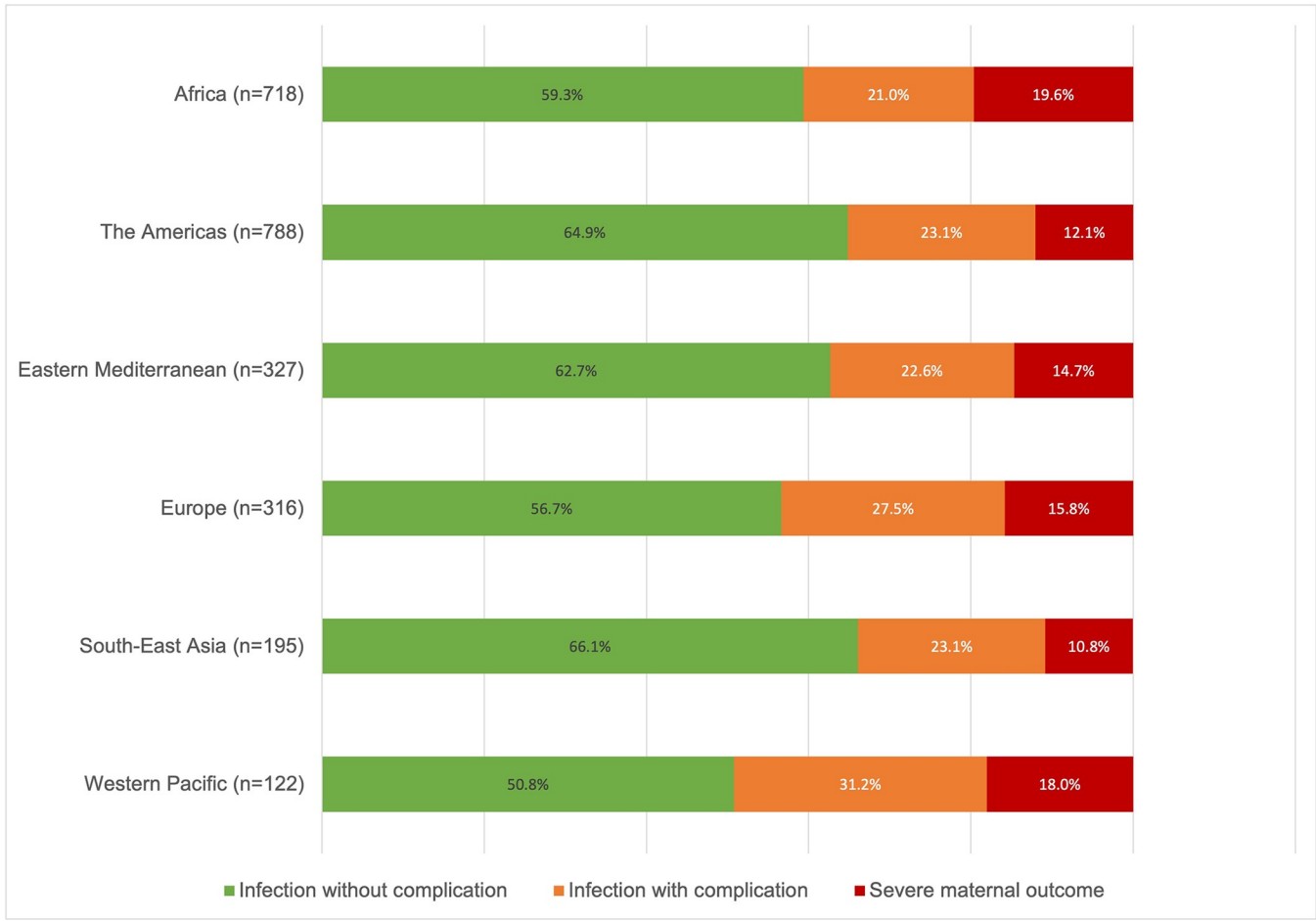

**Fig 1. Infection severity level by region.**

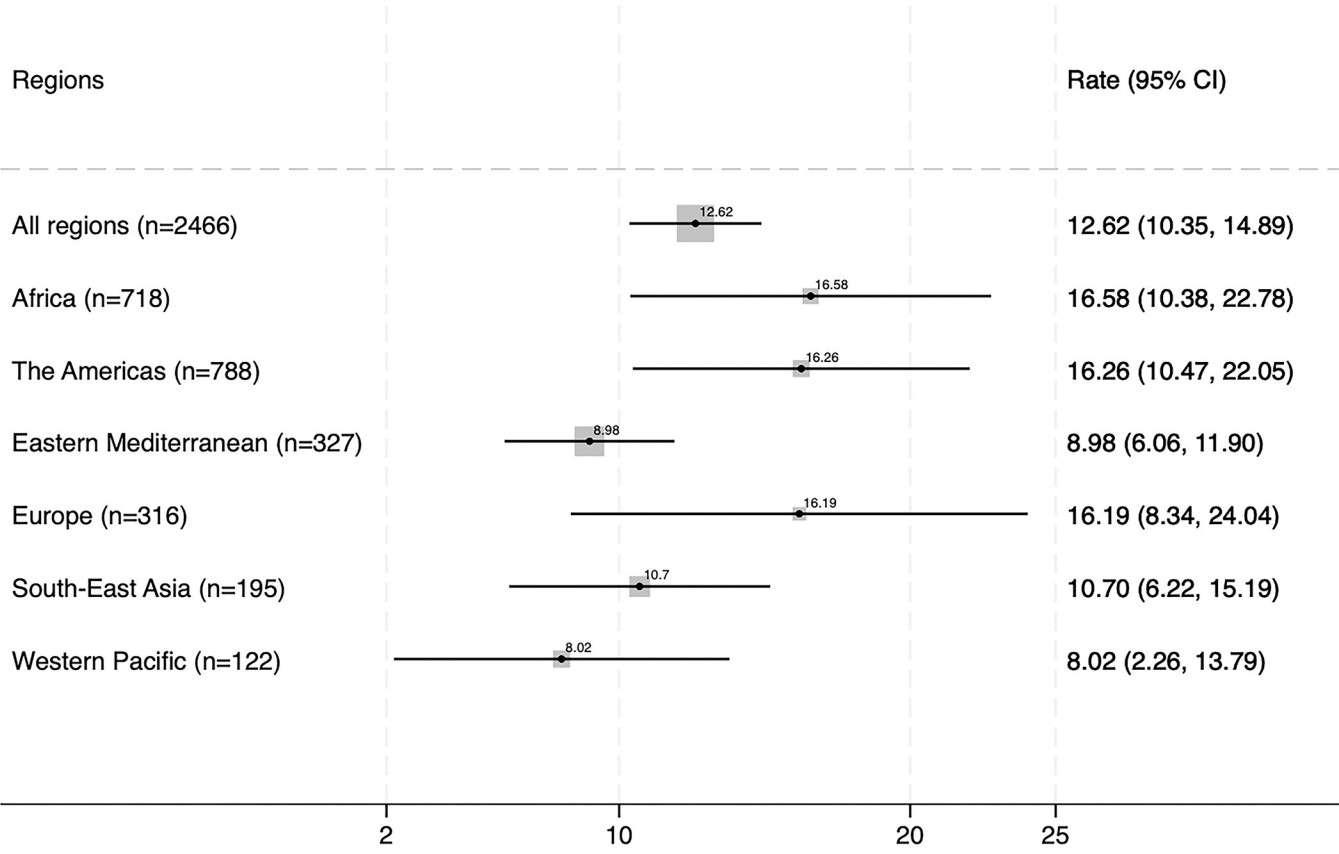

**Fig 2. Infection-related severe maternal outcomes rate per 1000 live births by region.**

Fig 2 shows that the rates of SMO per 1000 live births were also higher in Africa (16.58; 95%CI:10.38 to 22.78), the Americas (16.26; 95%CI:10.47 to 22.05), and Europe (16.19; 95% CI:8.34 to 24.04).

The forest plot in Fig 3 shows the overall and regional SMO fatality rate in percentage. In total, 26 women died, and the global case fatality rate was 6.9% (n/N = 26/377). In Africa and South-East Asia, it was 11.4% (n/N = 16/141) and 14.3% (n/N = 3/21) respectively.

## Association between the region and the occurrence of severe maternal outcome

We fitted a logistic regression model with women as units of analysis to explore the difference in the likelihood of experiencing a severe maternal outcome in Africa compared to the other five WHO regions. The results are presented in Table 3.

The unadjusted model showed that women in Africa were more likely to experience severe maternal outcomes than those living in the Americas (OR = 1.78, 95%CI: [1.34 to 2.36]) and in Southeast Asia (OR = 2.02, 95%CI: [1.24, 3.30]). After adjustment, women in Africa were more likely to experience SMO than those in the Americas (aOR = 2.41, 95%CI: [1.78 to 2.83]), the Eastern Mediterranean (aOR = 1.58, 95%CI: [1.08 to 2.32]), and South-East Asia (aOR = 2.60, 95%CI: [1.57 to 4.32]).

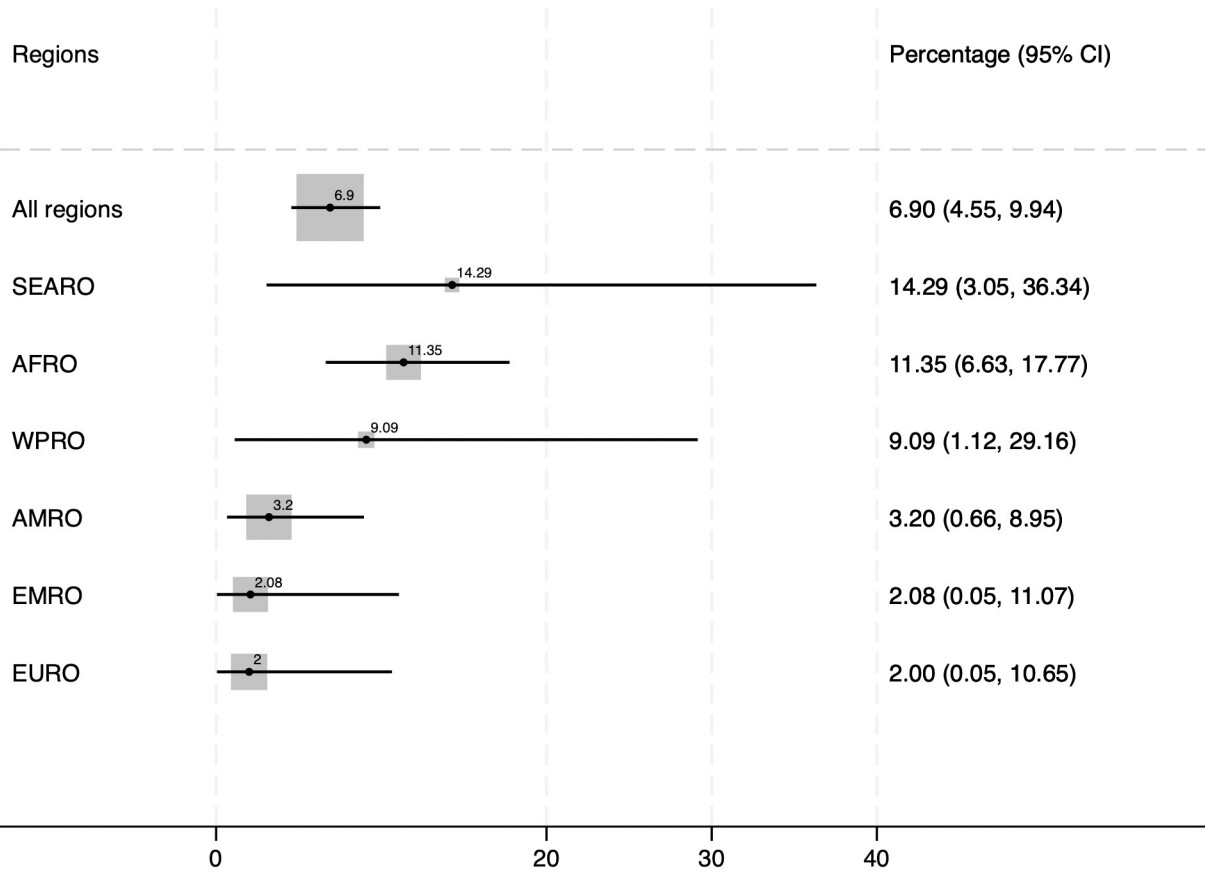

**Fig 3. Case fatality rate (%) among severe maternal outcomes by region.**

## Discussion

This analysis highlights differences in infection-related SMOs and case fatality rates across the WHO regions, using data from 43 LMICs of the GLOSS. One-fifth of women hospitalized with a suspected or confirmed infection in Africa experienced an SMO, the highest proportion among all the regions. Case fatality rates were also higher in South-East Asia and Africa. In addition, hospitals' readiness to identify and manage maternal infections was relatively lower in Africa than in the other regions.

**Table 3. Unadjusted and adjusted odds ratios (OR) to measure the association between the region and infection-related severe maternal outcome, comparing Africa to the other regions.**

| WHO region | OR | 95%CI | p-value | aOR* | 95%CI | p-value |
|---|---|---|---|---|---|---|
| Africa compared to the Americas | 1.78 | (1.34 to 2.36) | 0.000 | 2.41 | (1.78 to 2.83) | <0.001 |
| Africa compared to South-East Asia | 2.02 | (1.24 to 3.30) | 0.005 | 2.60 | (1.57 to 4.32) | <0.001 |
| Africa compared to the Eastern medditerranean | 1.42 | (0.99 to 2.03) | 0.054 | 1.58 | (1.08 to 2.32) | 0.018 |
| Africa compared to Europe | 1.30 | (0.91 to 1.85) | 0.146 | 1.08 | (0.74 to 1.58) | 0.677 |
| Africa compared to the Western Pacific | 1.11 | (0.68 to 1.83) | 0.679 | 1.02 | (0.61 to 1.68) | 0.953 |

*aOR = Adjusted odds ratio: For the facility size, location, capacity to manage maternal infections, and the availability of a routine training program on infection management, with robust variance.

This is the first analysis with global primary data that used standardized definitions related to maternal infectious complications across such a large number of hospitals. It adds to the previous analysis of the GLOSS [5,17] and the AMANHI study [2,4].

The low level of equipment, availability, and capacity for identification and the management of maternal infection in Africa is somehow related to the high incidence of SMO. Previous authors warned that some sepsis-related guidelines could not be implemented in sub-Saharan Africa due to the low availability of required facilities, equipment, and drugs [28]. Previous GLOSS analysis on the availability of resources and services by Brizuela et al. found differences in the availability of certain practices and resources across the country's income level. The lowest level of availability was reported in the group of low-income countries to which most African countries belonged [17]. Hospital management and women's clinical profile at admission are determinants of the burden of SMO in the regions. The differences in the proportion of SMOs across the regions could reflect low quality of care and weaknesses in health systems management [20,29].

All the global estimates indicate that sub-Saharan Africa and Asia (Central and South) have the highest maternal mortality ratios. But sub-Africa alone bears 70% of the global number of maternal deaths. With 545 deaths per 100,000 live births, the region has the highest maternal mortality ratio compared to any other region of the world, including South-East Asia [18]. Although the difference was not statistically significant, in our findings, the infection-related fatality rate was higher in South-East Asia than in Africa, which is quite surprising. There were contradictory findings in the literature in this regard. Some studies have shown that the proportion of maternal deaths attributable to maternal infections is higher in South Asia than in sub-Saharan Africa [2,4]. However, Chen et al. reported in their global estimates of the burden and trends of maternal sepsis and other maternal infections, that maternal mortality ratio in sub-Saharan Africa ranged from 19.40 (Western sub-Saharan Africa) to 71.54 (Central sub-Saharan Africa) deaths per 100 000 live births, and in Asia from 0.38 (East Asia) to 8.37 (South Asia) deaths per 100 000 live births [14]. There are differences in the definition and the measurements of these estimates. Still, the findings in our study could imply that the distribution of cause-specific (infection in particular) maternal mortality across the regions may differ from the overall mortality figure [30]. Therefore, assigning a death to a single cause can be problematic when weighing the importance of other concurrent causes [31]. Future studies on causes of maternal deaths, especially for maternal sepsis across the regions, could consider a more robust sample to explore differences in infection-related fatality rates across the regions because our study was somehow limited by the number of deaths identified during the follow-up.

Nonetheless, across all regions, preventing maternal infection-related morbidity and mortality will require further and specific actions. A critical aspect of prevention and management will be to improve hospitals' capacity to identify and manage infections, quality of care, and health system governance. The global community can assist countries in reviewing and improving their identification and treatment capacity [32], improving maternal infection surveillance, and investing in skilled health workers' availability, equipment, and supply [10,33,34].

This study has some limitations. First, the included facilities may not be fully representative of the regions. Second, the generalisability of the GLOSS findings is limited to intra-hospital outcomes and geographical areas similar to those included in the study. Third, our sample size was not powered to compare death rates; therefore, we could not accurately explore the association between the region and death rates. In addition, the availability of the services and equipment and the hospital's capacity to identify and manage maternal infections were based on the hospital's managers' self-reports. Therefore, these measures may be overestimated to show

better performance. However, this cohort study included participants prospectively. They were evaluated by the hospital's medical staff. In addition, the study was supported by an awareness campaign that helped improve the identification of the case and reduce the measurement and selection biases.

## Conclusion

This study showed disparities between Africa and the other regions regarding maternal infection-related SMO to the disadvantage of Africa. The hospitals in this region are relatively less equipped compared to others. The increased utilization of health services in recent years continually promoted through Universal Health Coverage, is an opportunity to treat complications and avert preventable maternal deaths, such as deaths due to infections. Therefore, the quality of in-hospital care is the main driver of women's and newborns' survival. Hospitals should, thus, be prepared for the prevention, early identification, and treatment of maternal infections with any severity level to significantly reduce the burden of severe maternal outcomes and intra-hospital fatality rate, particularly in the most disadvantaged regions.

## Supporting information

**S1 Fig. Percentage (%) of women with infection-related severe maternal outcome by region.**
(TIF)

**S1 Table. List of binary variables used to compute the composite variables.**
(DOCX)

**S2 Table. Services and equipment available in the facility.**
(DOCX)

**S3 Table. Identification and diagnostic capacity.**
(DOCX)

**S4 Table. Infection management-related characteristics of the facilities by region.**
(DOCX)

**S1 Acknowledgments.**
(DOCX)

## Acknowledgments

We sincerely thank the women who participated in this study. WHO is grateful to the extensive network of institutions and individuals who contributed to the project design and implementation, including researchers, study coordinators, data collectors, data clerks, and other partners, including the staff from the Ministries of Health and WHO offices. We would like to acknowledge the contribution and lifelong achievements of our late colleague Bukola Fawole, who passed away before the publication of this Article.

## Author Contributions

**Conceptualization:** Adama Baguiya, Mercedes Bonet.

**Data curation:** Adama Baguiya.

**Formal analysis:** Adama Baguiya.

**Methodology:** Adama Baguiya, Mercedes Bonet, Vanessa Brizuela.

**Writing – original draft:** Adama Baguiya.

**Writing – review & editing:** Mercedes Bonet, Vanessa Brizuela, Cristina Cuesta, Marian Knight, Pisake Lumbiganon, Edgardo Abalos, Séni Kouanda.

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
