## [Decision Letter · Decision Letter 0]

9 Feb 2024

PGPH-D-23-02196

Infection-related severe maternal outcomes and case fatality rates in 43 Low and Middle-Income Countries across the WHO regions: results from the Global Maternal Sepsis Study (GLOSS)

Dear Dr. Baguiya,

Thank you for submitting your manuscript to PLOS Global Public Health. After careful consideration, we feel that it has merit but does not fully meet PLOS Global Public Health’s publication criteria as it currently stands. Therefore, we invite you to submit a revised version of the manuscript that addresses the points raised during the review process.

We look forward to receiving your revised manuscript.

Kind regards,

Hannah Tappis, DrPH, MPH

Academic Editor

Journal Requirements:

1. In the online submission form, you indicated that "The data used for this analysis can be made available upon request. Contact of the GLOSS coordinator bonetm@who.int."

3. Uploaded as supplementary information.

Additional Editor Comments (if provided):

Reviewers' comments:

Reviewer's Responses to Questions

**Comments to the Author**

1. Does this manuscript meet PLOS Global Public Health’s publication criteria? Is the manuscript technically sound, and do the data support the conclusions? The manuscript must describe methodologically and ethically rigorous research with conclusions that are appropriately drawn based on the data presented.

Reviewer #1: Yes

Reviewer #2: Yes

Reviewer #3: Yes

2. Has the statistical analysis been performed appropriately and rigorously?

Reviewer #1: Yes

Reviewer #2: Yes

Reviewer #3: I don't know

3. Have the authors made all data underlying the findings in their manuscript fully available (please refer to the Data Availability Statement at the start of the manuscript PDF file)?

Reviewer #1: Yes

Reviewer #2: No

Reviewer #3: Yes

4. Is the manuscript presented in an intelligible fashion and written in standard English?

Reviewer #1: Yes

Reviewer #2: No

Reviewer #3: Yes

5. Review Comments to the Author

Reviewer #1: I read the article with big interest and I have nothing to add. The article is written very clearly, methodology is well described, results are completely illustrated in tables and figures, limits of the study are explained and conclusions are clear.

Reviewer #2: Dear authors and co-authors, thank you for reviewing an interesting manuscript.

Please see the specified questions and suggestions below.

Complete manuscript:

The manuscript should be checked by an English native speaker, there are unnecessary errors obscuring the quality of the study.

Abstract:

Lines 31-34: "The highest toll of maternal mortality due to infections is reported in low- and middle-income countries (LMICs). However, more evidence is needed to understand the differences related to severe maternal outcomes (SMO) and fatality rates across the WHO regions."

I would suggest adding the term “due to infections” to make it clear to the reader that the study only concerns SMO and fatality rates due to infections.

Line 40: "SMOs were infection-related maternal deaths or near-miss."

Should be changed to "SMOs are defined as maternal deaths and maternal near miss."

Sentences 41-42: The meaning is not clear, what does “on a set of countries and facilities characteristics” mean? Please clarify.

Lines 44-51: The results seem haphazardly presented, as a reader I would expect to find summarized results (Africa versus the other regions combined for example). Please consider revising.

Lines 62-64: I would expect some kind of comparison to be able to judge the percentage incidence of direct obstetric infection in sub-Saharan Africa. Standing alone, it is difficult to judge.

Lines 67-69: It comes across as outdated to use the term “developed” in this sentence, and in other places in the manuscript, especially since the rest of the manuscript uses the term LMIC.

Lines 72-74: The meaning of the sentence is not clear, does direct obstetric infections account for 11% of all maternal deaths, while in sub-Saharan Africa they constitute 12%? The next sentence also needs rephrasing, do you mean the third most common cause of maternal mortality?

Line 98: Maternal mortality is a ratio, not a rate, I would suggest to change to be precise.

Lines 126-127: It is not clear to the reader if Lithuania and Uruguay are considered high-income countries or were excluded for other reasons. Please clarify.

Lines 173-174: Did you aim to create three groups of equal size? If so, why?

Lines 184-185: It should be specified that you refer to the total number of women with SMO due to infection, not SMO overall.

Lines 196-200: If no identifiable information about individual women was entered into the system, it could not have been possible for them to request the withdrawal of their data. Please rephrase to what correct.

Lines 205-206: It is common practice to name the most frequent observation first, in this case the Americas before Africa.

Lines 206-207: I cannot find the number 977 in Table 1, do you mean Women who had a maternal infection (n=2466)? I cannot see a separate column for women with infection with complication.

Line 240: Table 2: I would suggest to put % in italics to increase readability.

Lines 261-263: It is unclear to me what S5 Figure depicts as there is no legend to the figure.

Lines 283: Table 2 includes unadjusted OR, this step is not mentioned in the methods section. Please revise.

Lines 292-293: It is more common, and accepted, not to report p-values among results due to its narrow interpretation, and misinterpretation. I would suggest to remove the sentence or rephrase.

Line 303: Be consistent, SMO is used in the rest of the manuscript and should be used also here.

Lines 348-350: The authors should phrase this finding with great caution since the number of maternal deaths in Southeast Asia only amounted to 3 in the study period. The short duration of the study and the extremely rare outcome makes it difficult to compare, and this caution should be communicated clearly to the reader.

Lines 373-377: The sentence is long and complicated, please consider revising.

Reviewer #3: This is a subanalysis of GLOSS study to evaluate certain regions and compare resources and outcomes and report disparities. My major concern is that the comparisons may not be valid based on the methods (see below). Descriptive info is very important and can highlight why certain regions have a greater proportion of case fatality rates and infection rates. The following is my suggested revisions to enhance the article:

Intro: lines 60-61 please use the WHO def of maternal sepsis rather than this layperson definition

lines 66-79, please consider arranging by infection on a global level, then focus on the specifics to sub-Saharan Africa and South Asia for clarity.

Consider shortening the intro as it is lengthy.

Methods: lines 180-189, can you please comment on how the final model was adjusted? Where all variable treated equally? Or where there adjustments based on how the income affected the resources, etc.? I would think that the lower the income, the more that might affect the capacity of the hospital to identify and treat infection.

Table 1: Please consider just spelling out the abbreviations. It's hard to follow even with the key at the bottom.

Table 2. There is a lot of variability in the number of facilities in each region. How was this managed?

Table 3: These comparisons are not particularly helpful since the regions are not equal and I am not sure the model adequately adjusts for the differences.

Discussion: this should be shortened significantly.

I think this paper is a great addition to the literature that gives information on where to invest to improve infection in maternity care.

6. PLOS authors have the option to publish the peer review history of their article (what does this mean?). If published, this will include your full peer review and any attached files.

**Do you want your identity to be public for this peer review?** For information about this choice, including consent withdrawal, please see our Privacy Policy.

Reviewer #1: No

Reviewer #2: No

Reviewer #3: No

---

## [Editor Report · Decision Letter 1]

1 Apr 2024

Infection-related severe maternal outcomes and case fatality rates in 43 Low and Middle-Income Countries across the WHO regions: results from the Global Maternal Sepsis Study (GLOSS)

PGPH-D-23-02196R1

Dear Dr. Baguiya,

We are pleased to inform you that your manuscript 'Infection-related severe maternal outcomes and case fatality rates in 43 Low and Middle-Income Countries across the WHO regions: results from the Global Maternal Sepsis Study (GLOSS)' has been provisionally accepted for publication in PLOS Global Public Health.

Best regards,

Hannah Tappis, DrPH, MPH

Academic Editor